# PCSK9 Inhibition: From Current Advances to Evolving Future

**DOI:** 10.3390/cells11192972

**Published:** 2022-09-23

**Authors:** Chunping Liu, Jing Chen, Huiqi Chen, Tong Zhang, Dongyue He, Qiyuan Luo, Jiaxin Chi, Zebin Hong, Yizhong Liao, Shihui Zhang, Qizhe Wu, Huan Cen, Guangzhong Chen, Jinxin Li, Lei Wang

**Affiliations:** 1State Key Laboratory of Dampness Syndrome of Chinese Medicine, The Second Affiliated Hospital of Guangzhou University of Chinese Medicine, Guangzhou 510080, China; 2Guangdong-Hong Kong-Macau Joint Lab on Chinese Medicine and Immune Disease Research, Guangzhou 510080, China; 3State Key Laboratory of Quality Research in Chinese Medicine, Institute of Chinese Medical Sciences, University of Macau, Macau 999078, China; 4School of Biotechnology and Health Sciences, Wuyi University, Jiangmen 529020, China; 5Health Science Center, Shenzhen University, Shenzhen 518060, China; 6Department of Neurosurgery, Institute of Neuroscience, Guangdong Provincial People’s Hospital, Guangdong Academy of Medical Sciences, Guangzhou 510080, China

**Keywords:** cardiovascular disease, PCSK9 inhibitors, clinical applications, security, research status

## Abstract

Proprotein convertase subtilisin/kexin type 9 (PCSK9) is a secretory serine protease synthesized primarily by the liver. It mainly promotes the degradation of low-density lipoprotein receptor (LDL-R) by binding LDL-R, reducing low-density lipoprotein cholesterol (LDL-C) clearance. In addition to regulating LDL-R, PCSK9 inhibitors can also bind Toll-like receptors (TLRs), scavenger receptor B (SR-B/CD36), low-density lipoprotein receptor-related protein 1 (LRP1), apolipoprotein E receptor-2 (ApoER2) and very-low-density lipoprotein receptor (VLDL-R) reducing the lipoprotein concentration and slowing thrombosis. In addition to cardiovascular diseases, PCSK9 is also used in pancreatic cancer, sepsis, and Parkinson’s disease. Currently marketed PCSK9 inhibitors include alirocumab, evolocumab, and inclisiran, as well as small molecules, nucleic acid drugs, and vaccines under development. This review systematically summarized the application, preclinical studies, safety, mechanism of action, and latest research progress of PCSK9 inhibitors, aiming to provide ideas for the drug research and development and the clinical application of PCSK9 in cardiovascular diseases and expand its application in other diseases.

## 1. Introduction

Proprotein convertase subtilisin/kexin type 9 (PCSK9) is the ninth member of the subtilisin protease family. In 2003, it was discovered in the brain by the Montreal Clinical Research Institute, Canada, and was named neural apoptosis-regulated convertase 1 (NARC-1) before being modified to PCSK9 according to the standard nominating method [1]. PCSK9 protein is formed by the connection of signal peptide, proregion, catalytic region, and carboxyl-terminal [2], is located on human chromosome 1P32, close to the third genetic site, and is mainly expressed in the liver, kidney, and intestine [3]. As a secreted protein, PCSK9 is similar to other proto-protein invertases, it is secreted into the circulation as a heterodimer after intramolecular autocatalytic cleavage of a precursor protein in the liver. It then binds to and internalizes EGF-A, the epidermal growth factor precursor domain of low-density lipoprotein receptor (LDL-R) [4], regulating its circulation on the surface of hepatocytes to manage the low-density lipoprotein cholesterol (LDL-C) concentration. The affinity between LDL-R and PCSK9 increases under acidic conditions, and the conformation changes after binding, which prevents normal circulation and the ability to enter lysosomes for degradation in the form of the LDL-R-PCSK9 complex. In addition to LDL-R, PCSK9 also binds to Toll-like receptors (TLRs) to mediate inflammatory responses and SR-B receptors to promote platelet activation (PA) and thrombosis. Low-density lipoprotein receptor-related protein 1 (LRP1), apolipoprotein E receptor-2 (ApoER2), very-low-density lipoprotein receptor (VLDL-R), and other receptors can promote vascular endothelial hyperplasia and increase lipoprotein concentrations.

PCSK9 inhibitors are mainly used in clinical hyperlipidemia, atherosclerosis (As), and related ischemic cardiovascular diseases with significant effects and can be used as a substitute for statins and ezetimibe [5,6,7,8]. PCSK9 inhibitors can inhibit vascular endothelial cell injury, delay plaque formation and protect myocardial cells [9,10,11,12]. PCSK9 is involved in the occurrence and development of cardiovascular diseases through a variety of mechanisms, such as LDL-R degradation mediating the increase in circulating low-density lipoprotein (LDL) [4,13], binding with TLRs mediating the inflammatory response [14,15,16,17] and binding with the CD36 receptor promoting PA and thrombosis [18,19]. In addition, lipid-lowering is not the only function of PCSK9 inhibitors. Existing studies suggest that PCSK9 inhibitors have potential application value in the treatment of sepsis, some tumors, some viral infections, and other diseases [20,21,22,23,24]. Whether they can bring more clinical benefits needs to be confirmed by further studies.

Since the first PCSK9 inhibitor was developed in 2015, it has been rapidly applied in clinical practice with a strong lipid-lowering effect and good safety and tolerability. Currently, the main PCSK9 inhibitors used in the clinic are alirocumab, evolocumab, and inclisiran. Although these drugs have been proven to be safe after clinical use, adverse reactions such as nasopharyngitis and mild self-limited injection site reactions persist [25,26]. In addition, the use of PCSK9 inhibitors is associated with a significant decrease in plasma LDL-C levels and the risk of cognitive impairment (such as delirium, attention disorder, amnesia, dementia, thinking and perception impairment, or mental disorder) [27]. However, the conclusion has been disproved by randomized clinical trials [28,29,30]. Based on these trials, very low LDL-C is safe and not associated with cognitive issues.

This paper reviewed the mechanism of PCSK9, the classification and clinical application of existing PCSK9 inhibitors, and the latest progress in PCSK9-related research, aiming to provide a reference for clinical drug use and later drug development.

## 2. Molecular Mechanism of PCSK9

PCSK9, a member of the subtilis protease K subfamily, encodes the preprotein invertase Bacillus subtilis protein, which is released into the peripheral circulation after autocatalytic maturation in the endoplasmic reticulum and binds with different receptors or molecules to produce different biological effects. The mechanisms of PCSK9 action can be divided into six categories: binding to LDL-R to degrade LDL, binding to TLRs to mediate the inflammatory response, binding to CD36 receptor to promote PA and thrombosis, and binding to LRP1, ApoER2, and VLDL-R to promote lipoprotein concentration. This section summarizes the molecular mechanism of PCSK9 action, which is expected to provide an important basis and guidance for drug research and development (Figure 1).

### 2.1. PCSK9 Increases Circulating LDL-C by Degrading LDL-R

PCSK9 targets LDL-R degradation [31,32], which is an important target for reducing LDL-C in blood circulation [28,33]. LDL-R is crucial to the metabolism of LDL particles. LDL-C combines with LDL-R on the surface of the liver cell membrane and enters human liver cells through endocytosis. A decrease in the intracellular pH value will separate LDLC from LDL-R. After separation, LDL-C is degraded in the lysosome, and LDL-R returns to the surface of the liver membrane to continue to bind the remaining LDL-C. After PCSK9 binds with LDL-R, the PCSK9-LDL-R-LDL complex is formed, which causes PCSK9, LDL-C, and LDL-R to enter the lysosome together to be degraded, LDL-R on the cell surface decreases, and LDL-C degradation decreases accordingly [34,35]. Based on this process, studies have shown that the PCSK9-mediated increase in circulating LDL is closely related to the progression of cardiovascular diseases such as coronary heart disease [36,37].

A meta-analysis evaluating a large sample of patients (*n* = 28,319) found a significant positive association between circulating PCSK9 concentration and the risk of major adverse cardiovascular events [38], suggesting that inhibition of PCSK9 expression reduces serum LDL levels and the risk of cardiovascular disease [5,39]. In addition, the level of circulating PCSK9 protein is independent of known risk factors, including LDL, and can be used to predict future cardiovascular events [6]. Intravascular ultrasound virtual histology imaging showed a linear correlation between serum PCSK9 levels and the proportion and amount of necrotic core tissue in coronary As but not with serum LDL-C levels and statin use [7]. Similarly, in ApoE^−/−^ mice, overexpression of PCSK9 increased plaque size in the aortic sinus and aortic root without changing plasma cholesterol levels [5,11]. This suggests that PCSK9 may also influence vascular biology and the progression of cardiovascular disease through other mechanisms.

### 2.2. PCSK9 Binds to TLR4 to Mediate the Inflammatory Response

TLRs are transmembrane proteins defined by cytoplasmic domains for ligand recognition of extracellular domains and interactions with TLR signal transduction proteins that activate downstream NF-KB pathways of biological signals. PCSK9 can bind to TLRs, increase the levels of P-IκBα, IkBα degradation, and NF-κB nuclear translocation in macrophages, regulate the microenvironment of myocardial inflammation, and influence the course of cardiovascular disease.

PCSK9 increases the secretion of inflammatory factors mainly by promoting the activation of the TLR4/NF-KB pathway, and the exact mechanism may be related to the similarity of resistin and the C-terminus of the cysteine-rich domain of the PCSK9 protein. Resistin binds to TLR-4 through the C-terminus and upregulates its expression to activate the TLR-4 signaling pathway, suggesting that PCSK9 and resistin have similar effects on TLR-4 [9]. Tissue factor, a glycoprotein that plays important roles in coagulation and inflammation, is rapidly induced by proinflammatory agents through NF-KB-dependent mechanisms stimulating circulating monocytes. Valentina Scalise et al. found that PCSK9 activated the TLR4/NF-KB signaling pathway in TLR4-HEK293 cells to the same extent as lipopolysaccharide (LPS), and the colocalization of PCSK9 and TLR4 was confirmed by TLR4-specific agonists and quantitative confocal microscopy [40]. This suggests that PCSK9 can activate the TLR4/NF-KB signaling pathway to induce tissue factor expression, which was independent of the proinflammatory effect of circulating LDL-C. The injury and activation of endothelial cells may be the core link with the occurrence and development of sepsis. The content of endothelial cell microparticles can be significantly increased after sepsis and affect endothelial function by regulating nitric oxide, nitric oxide oxygenase, reactive oxygen species, and other signaling molecules, promoting inflammation and microvascular injury. Longxiang Huang et al. found that in septic mice, the expression levels of eNOS and VE-cadherin were decreased, and the expression of PCSK9 was increased. Activation of the TLR4/MyD88/NF-κB and NLRP3 pathways was responsible for the endothelial dysfunction induced by PCSK9. Inhibition of PCSK9 can prevent the decline of endothelium-dependent vasodilation and improve the survival rate of septic mice [41]. These results suggest that increased PCSK9 in sepsis activates the TLR4/MyD88/NF-κB and NLRP3 pathways to induce inflammation, resulting in vascular endothelial dysfunction and a reduced survival rate. In conclusion, PCSK9 can combine with TLR4 to regulate the expression of inflammatory factors and affect the process of inflammatory diseases.

### 2.3. PCSK9 Binds to CD36 to Promote Platelet Activation and Thrombosis

CD36 exists in platelets, mononuclear phagocytes, and adipocytes, and acts as a negative regulator of angiogenesis. As scavenger receptors on phagocytic membranes, CD36 participates in the internalization of apoptotic cells, bacterial and fungal pathogens, and LDL by recognizing specific oxidized phospholipids and lipoproteins, mediating inflammatory responses that affect the progression of As. In addition, PCSK9 can also bind to platelet CD36, enhance PA, and reduce fatty acid uptake and triglyceride accumulation in tissues. Circulating PCSK9 is known to enhance platelet activation (PA) and PCSK9i to decrease it, but the underlying mechanism is unknown. Vittoria Cammisotto et al. conducted a multicenter before and after control study on 80 heterozygotes. Patients with familial hypercholesterolemia (HeFH) receiving the maximally tolerated statin dose ± ezetimibe, PA, soluble Nox2-derived peptide (SNOX2-DP) and oxidized low-density lipoprotein (ox-LDL) were measured before and after 6 months of PCSK9i treatment. Compared with the baseline, PCSK9i decreased serum LDL-c, ox LDL, thrombosis, alkane (Tx) B2, sNOX2-dp and PCSK9 levels (*p* < 0.001). These results suggest that PCSK9i treatment can sequentially reduce ox-LDL formation in PA-regulated NOX2 active HeFH patients [42].

The involvement of PCSK9 in PA has been demonstrated in animal models of PCSK9^−/−^ mice [43], but a clear association between PCSK9 levels and platelet reactivity is still lacking [18]. Recent studies have shown that PCSK9 protein in plasma binds to the CD36 receptor and activates Src kinase and mitogen-activated protein kinase (MAPK) extracellular signal-regulated kinase 5 and C-Jun amino-terminal kinase, upregulating ROS levels. In addition, the p38MAPK/cytoplasmic phospholipase A2/cysin-1/thromboxane A2 signaling pathway downstream of CD36 is activated to promote PA and thrombosis in vivo [19]. The formation of arterial circulation thrombosis is the main pathological cause of arterial thrombotic diseases such as acute coronary syndrome and ischemic stroke, and PA, aggregation of vascular injury sites, and subsequent thrombosis are the key steps of these arterial thrombotic diseases [44].

In addition, PA also aggravates microvascular obstruction and promotes post-MI dilation [19]. PCSK9 is involved in triglyceride metabolism, independent of its effect on LDL receptors, and has been suggested to be mediated by CD36. Annie Demers et al., induced CD36 degradation in cell lines and primary adipocytes through overexpression of PCSK9 and reduced the uptake of the palmitate analog BODIPY FL C16 and OX-LDL in 3T3-L1 adipocytes and HepG2 hepatocytes, respectively. Other studies found that the level of CD36 protein significantly increased the expression of small interfering RNA (siRNA) combined with endogenous PCSK9 in liver cells and liver and visceral adipose tissues of PCSK9^−/−^ mice [45], suggesting that PCSK9 plays an important role in regulating CD36 and triglyceride metabolism.

In conclusion, PCSK9 promotes PA and thrombosis by binding to CD36 and participates in triglyceride metabolism, limiting fatty acid uptake and triglyceride accumulation in liver tissues.

### 2.4. PCSK9 Binds to Lipid-Associated Receptors to Regulate Metabolism

Mutations in PCSK9, a third locus associated with familial hypercholesterolemia, lead to a higher clearance of plasma LDL-C owing to reduced degradation of liver LDL-R. In addition, studies have shown that the two family members closest to LDL-R, VLDL-R, and ApoER2, are also targets of PCSK9 and are mainly involved in neuronal development and lipid metabolism.

LDL-R is the primary regulator of circulating LDL levels, and VLDL-R and ApoER2 in the brain mediate Reelin signaling, a key pathway for normal nervous system development. Steve Poirier et al. found that wild-type PCSK9 and its naturally acquired functional mutation D374Y can effectively degrade LDL-R, VLDL-R, and ApoER2 after cell coexpression or reinternalization of secreted human PCSK9 without requiring catalytic activity for induced degradation. Membrane-bound PCSK9 chimera enhances the intracellular targeting of late endosomes/lysosomes. This study also demonstrated that the activity of PCSK9 and its binding affinity for VLDL-R and ApoER2 was independent of the presence of LDL-R, while the expression of PCSK9 mRNA was closely related to VLDL-R expression in the cerebellums of suckling mice. This demonstrates a more general effect of PCSK9 on the degradation of the LDL-R family, highlighting its major roles in cholesterol and lipid homeostasis and brain development [46].

The expression level of PCSK9 in the brain was highest in the perinatal cerebellum but also increased in the adult brain tissue after ischemia. The function of PCSK9 and the mechanism of its involvement in neuronal apoptosis remain unclear. Kai Kysenius et al. found that PCSK9 KO significantly reduced the death of potassium-deficient cerebellar granular neurons, as demonstrated by decreased levels of nuclear-phosphorylated C-Jun, activated caspase-3, and apoptotic nucleus concentrations. Knockdown of ApoER2 is insufficient to reverse the protective effect provided by PCSK9 RNAi, suggesting that the preapoptotic signaling pathway of PCSK9 is mediated by changes in ApoER2 function. Studies have shown that PCSK9 regulates neuronal apoptosis independently of NMDA receptor function but synergistically with the ERK and JNK signaling pathways [47]. These results suggest that PCSK9 enhances neuronal apoptosis by regulating ApoER2 levels and related antiapoptotic signaling pathways. In addition, it has been found that PCSK9 in the brain can promote neuronal apoptosis by activating the Bcl-2/Bax/Caspase3 signaling pathway. A recent study also found that the level of PCSK9 in the cerebrospinal fluid of AD patients was higher than that of non-AD patients, indicating that there is a pathophysiological link between PCSK9 and AD, and PCSK9 may play a role in AD by activating lipid accumulation, apoptosis and amyloid production in the brain [48]. These findings suggest that PCSK9 has a more general effect on cellular degradation and that the LDL-R family emphasizes its roles in cholesterol and lipid homeostasis and brain development.

In conclusion, PCSK9 is involved in lipid metabolism, mediates the inflammatory response, promotes PA and thrombosis, and mediates neuronal apoptosis through the abovementioned mechanisms. PCSK9 has a wide range of effects and has become a potential therapeutic target for a variety of diseases. However, its more precise association with diseases needs to be further confirmed.

## 3. Classification of PCSK9 Inhibitors

PCSK9 inhibitors are mainly divided into four categories: (1) McAb inhibitors, such as alirocumab (Regeneron, New York, NY, USA; and Sanofi, Paris, France) and evolocumab (Amgen, Los Angeles, CA, USA). PCSK9 binds to LDL-R through direct inhibition, such as monoclonal antibodies and analog peptides [25]. PCSK9 monoclonal antibody blocks the interaction between PCSK9 and LDL-R inhibits the degradation process of LDL-R and reduces LDL-C levels. Analog peptides play an inhibitory role by blocking the binding of PCSK9 to the EGF-A domain of LDL-R. (2) Nucleic acid drugs, such as antisense oligonucleotides (ASOs), siRNAs, and CRISPR/Cas9 gene editing systems. ASOs bind to the mRNA of the target PCSK9 target gene through Watson-Crick base pair interactions and target gene expression is restricted [49]. siRNA interferes with mRNA degradation of specific sequences and inhibits PCSK9 gene expression, thereby inhibiting the synthesis of corresponding proteins [50]. CRISPR/Cas9 can reduce the expression of PCSK9 target genes by inducing host cell DNA double-strand breaks, resulting in error-prone structure, recombination repair, and nonhomologous terminal connection [51]. (3) Small molecules, such as BMS-962476. Small molecules block the biological activity of PCSK9 by preventing the binding of LDL-R, resulting in the disruption of subsequent sorting and degradation steps and increased receptor circulation and LDL uptake [52]. (4) Vaccine drugs, such as the L-IFPTA + vaccine, can inhibit circulating PCSK9 activity [53]. This section summarizes the current classification of PCSK9 inhibitors, which is expected to provide some basis and guidance for their development (Table 1).

### 3.1. McAb Inhibitors

PCSK9 is a secreted protein involved in regulating the life cycle of liver LDL-R. By inhibiting the circulation of PCSK9, monoclonal antibodies reduce the plasma level of LDL-C, which has benefited many patients, and their effectiveness in reducing LDL-C has also been proven in clinical trials in high-risk cardiovascular patients [67].

Currently, PCSK9 monoclonal antibodies include alirocumab and evolocumab, which are approved by the US Food and Drug Administration (FDA) for the treatment of hypercholesterolemia, including primary hypercholesterolemia and familial hypercholesterolemia. In a Rutherford-2 clinical trial in patients with heterozygous familial hypercholesterolemia (HeFH), evolocumab (140 mg/2 weeks or 420 mg/month) reduced LDL-C levels by 60 to 65%, and more than 60% of patients were able to achieve their LDL-C target of 70 mg/dL [68]. In the Odyssey High Clinical trial, 57% of patients with refractory severe familial hypercholesterolemia were able to achieve LDL-C levels of 100 mg/dL after treatment with alirocumab (75/150 mg every 2 weeks) [69].

In addition, alirocumab in combination with statins showed a significant reduction in LDL-C compared with patients who were given double the dose of statins alone or who received ezetimibe [25]. Alirocumab and evolocumab have been approved by the US FDA for HeFH and the prevention of CV events in patients with diagnosed cardiovascular disease. However, there is limited evidence for PCSK9 monoclonal antibody in homozygous familial hypercholesterolemia (HoFH), so only evolocumab has been labeled for this indication [33].

Excessive muscular symptoms or elevated liver enzyme levels are adverse symptoms of statin therapy, but no similar adverse effects have been found in PCSK9 monoclonal antibody studies thus far. Nasopharyngitis and mild self-limited injection site reactions (e.g., itching, redness, and swelling) are considered to be the most common adverse reactions of PCSK9 monoclonal antibody administration. In general, PCSK9 monoclonal antibody alone or in combination with statins is an ideal lipid-lowering agent with high efficiency and few adverse reactions.

### 3.2. Nucleic Acid Drugs

The nucleic acid drugs of PCSK9 inhibitors are mainly ASOs, which are designed to target PCSK9 mRNA and thereby inhibit intracellular protein translation and PCSK9 protein synthesis through the mechanism of occupying or cutting the targeted mRNA by RNase H1 [70].

The current PCSK9 nucleic acid inhibitor, inclisiran, developed by Novartis, is administered by subcutaneous injection. However, a common disadvantage of injecting drugs is adverse reactions at the injection site. Inclisiran also inevitably produced injection site reactions, pain, erythema, rash, and other symptoms during use. To avoid this situation and increase the convenience of drug use, the development of oral drugs is crucial. American medical researchers have developed the ASO drug AZD8233 (also known as ION-863633) with oral potential, which is a highly efficient 16-nucleotide oligomer, three-chain N-acetylgalactosamine (GalNAc) ASO that inhibits the chemistry of ethyl. This drug uses galnac-ASO to target the expression of endogenous liver PCSK9 to achieve the ablation of circulating PCSK9 and induce the expression of liver LDL-R, thereby reducing plasma total cholesterol and LDL-C concentrations. AZD8233 was confirmed to be feasible for oral administration by American physicians after tests on animal models such as dogs, mice, and cynomolgus monkeys [59]. However, existing studies on the liver bioavailability of AZD8233 at clinically relevant doses have only been performed in rats and dogs, and it is not clear how to apply these findings to humans. In addition, there is currently no clinical study to evaluate the use of a once-daily low-dose ASO and its effect on target cell exposure, so it has not been determined whether it can be used orally [57].

In addition to ASOs, miRNAs can also inhibit the generation of PCSK9. After screening and experimental verification of the TargetScan database, miR-552-3p was identified as a negative regulator of PCSK9. By regulating the expression of LDL-R on the membrane, miR-552-3p reduces the expression of PCSK9 in HepG2 cells and promotes the uptake of LDL-C by HepG2 cells, reducing the level of serum LDL-C. In addition, studies have shown that miR-552-3p could inhibit PCSK9 translation by binding the PCSK9 3′-UTR and could inhibit PCSK9 transcription by binding the PCSK9 promoter. By these two methods, the expression of PCSK9 mRNA was inhibited, the formation of PCSK9 was reduced, the expression of LDL-R was increased, and lipid abnormalities were significantly improved. These results suggest that the use of miR-552-3p may be a therapeutic target for hyperlipidemia and a more effective therapeutic option for inducing miR-552-3p [58].

### 3.3. Small-Molecule Drugs

The binding interface of PCSK9 protein is relatively open and flat, the contact area of small molecules is limited, and the hydrophobic force generated is weak. Even if it is bound to PCSK9 protein, it may be dissociated when touched by natural macromolecular ligands. Hence, there are some difficulties in the research and development of small molecule drugs [71], but some progress has been made. Adnectin and BMS-962476 with high affinity for PCSK9 were developed successively.

BMS-962476 is a pegylated polypeptide that enhances pharmacokinetics and binds with subnanomolar affinity in humans. Some studies established a genomic DNA model of human PCSK9 transgenic mice, and it was found in the model that BMS-962476 can rapidly reduce cholesterol and free PCSK9 levels. In addition, BMS-962476 was used to treat cyclamophagus monkeys, and PCSK9-free cyclamophagus monkey plasma was found to be rapidly inhibited.

At the same time, data from various cell-based PCSK9 inhibition tests showed that BMS-962476 blocked the biological activity of PCSK9 by preventing the binding and coassimilation of LDL-R during endocytosis, resulting in the disruption of subsequent sorting and degradation steps and increased receptor circulation and LDL uptake. These results suggest that BMS-962476 may be a highly effective inhibitor of PCSK9. It represents a new class of high-purity PCSK9 small-molecule inhibitors that exhibit many of the functional properties of inhibiting PCSK9 monoclonal antibodies but are suitable for polypeptides and immunoglobulins within the molecular size range. BMS-962476, as an effective PCSK9 inhibitor, may have the potential to treat LDL in patients with hypercholesterolemia in the treatment of cardiovascular diseases [52].

### 3.4. Vaccine Drugs

PCSK9 inhibitors, which are critical negative regulators of LDLR, have been a breakthrough in the field of lipid-lowering therapy. Monoclonal antibody drugs against PCSK9 have been widely used in the field of lipid-lowering, but their use has certain limitations, such as a short half-life in vivo, resulting in frequent administration and high cost. To circumvent these limitations, the PCSK9 vaccine may be a better choice [72].

In terms of vaccine development, some researchers constructed immunogenic peptide constructs fusing PCSK9-tetanus (IFPT) on the surface of liposome nanoparticles (L-IFPT) and then mixed them into alum adjuvant [73]. The L-IFPTA + vaccine had the highest IgG response to PCSK9, which induced the production of a PCSK9 antibody. The antibody directly targeted and eliminated circulating PCSK9 from the blood, which inhibited the PCSK9-LDL-R interaction and increased liver LDL-R protein [74]. In addition, the induced antibodies were persistent; the antibody titer was measured 16 weeks after the vaccine injection, and it was found that the antibody titer remained high. The L-IFPTA vaccine has the potential to be an effective alternative to monoclonal antibody-based therapies for controlling elevated cholesterol levels and preventing cardiovascular disease [60].

### 3.5. CRISPR/Cas9-Targeted KO Drugs

CRISPR/Cas9, a cluster of regularly spaced short palindromic repeats/CRISPR-associated protein 9, is a state-of-the-art genome editing technology that enables researchers to better target gene therapy for a wide range of diseases. Studies in recent years have found that drugs edited by CRISPR/Cas9 can target PCSK9 and thus reduce blood cholesterol levels. Although this drug is still in the basic research stage, various studies have shown that CRISPR/Cas9-targeted KO drugs have a very high potential in alleviating dyslipidemia [75].

CRISPR/Cas9-targeted drugs are mainly directed to specific genomic regions by their Cas9 nucleases and cleaved DNA target sequences with a single guide RNA (sgRNA). To achieve this, a matching delivery system is needed. Among them, adeno-associated virus (AAV)-mediated CRISPR–Cas9 system delivery showed high gene targeting efficacy in vivo, and AAV has low immunogenicity and serotype specificity, which can safely deliver the CRISPR–Cas9 system to a variety of cell types, tissues, and organs [76]. Ding, Q. et al., delivered a CRISPR/Cas9-targeted KO drug to mice and found that it could effectively destroy the PCSK9 gene in vivo, resulting in a decrease in circulating PCSK9 levels, an increase in liver LDL-R levels, and ultimately a decrease in blood cholesterol levels in mice, indicating that it also has the potential to be a PCSK9 inhibitor [77]. In a recent study, Sekar Kathiresan et al., used the single-base editing tool ABE and lipid nanoparticles (LNPs) to achieve precise editing of PCSK9 gene splicing sites in a nonhuman primate model. A single injection of the construct permanently inhibited the expression of the PCSK9 gene in the liver, resulting in an effective reduction in PCSK9 and LDL levels in the blood. This study applied CRISPR/Cas9 to PCSK9, providing a new idea for the prevention of cardiovascular diseases [78]. To date, CRISPR/Cas9-targeted drugs (VT-1001) have been tested in phase I clinical trials. VERVE-101 uses base-editing technology designed to disrupt the expression of the PCSK9 gene in the liver and lower circulating PCSK9 and LDL-C in patients with established ASCVD due to HeFH, which was used to evaluate the safety of VERVE-101 administered to patients with heterozygous familial hypercholesterolemia (HeFH), atherosclerotic cardiovascular disease (ASCVD), and uncontrolled hypercholesterolemia [79].

## 4. Clinical Application of PCSK9 Inhibitors

PCSK9 inhibitors have significant efficacy in hyperlipidemia, atherosclerosis, and other diseases, with many reports highlighting the potential therapeutic effects of PCSK9 inhibitors in treating sepsis, tumors, and some viral infections, among other diseases. This section will introduce their application in more detail (see Figure 2).

### 4.1. Hyperlipidemia

Blood lipids include cholesterol, triglycerides (TGs), and lipids in serum. Hyperlipidemia includes elevated serum total cholesterol (TC), LDL-C, and TG. The treatment of hyperlipidemia requires diet adjustment and adverse lifestyle changes regardless of drug therapy. Commonly used drugs include statins, cholesterol absorption inhibitors, bempedoic acid, high-purity fish oil preparations, and PCSK9 inhibitors [80]. Inhibition of PCSK9 is a new therapeutic strategy for the control of hyperlipidemia, which can improve LDL-R circulation, increase the utilization of LDL-R on the surface of liver cells, and reduce the level of LDL-C in the blood. A long-term study of patients with hypercholesterolemia using a monoclonal antibody [81] found that evolocumab reduced LDL-C concentrations by 55–57% compared with placebo. The study also compared evolocumab monotherapy with ezetimibe and found a 38 to 39% reduction in LDL-C in the evolocumab group, suggesting a significant reduction in LDL-C with a single PCSK9 inhibitor.

The mechanism by which statins reduce LDL-C levels is to increase LDL-R expression on the surface of liver cells, thereby reducing circulating LDL-C levels. However, the negative feedback regulation of statins induces the expression and secretion of PCSK9, which weakens its efficacy in lowering LDL [82]. Therefore, the combination of PCSK9 inhibitors may provide an effective way to reduce LDL-C in patients with drug resistance treated with statins alone. PCSK9 inhibitors were found to reduce circulating LDL-C in patients with HeFH, and the effect of ezetimibe combined with evolocumab was similar to that of evolocumab alone [68]. Another study found that the addition of evolocumab to hyperlipidemia resulted in an average 48% reduction in LDL-C compared with placebo [83]. The above experiments indicate that ezetimibe does not significantly affect the lipid-lowering efficacy of PSCK9 inhibitors, and further lipid lowering can be achieved based on the efficacy of ezetimibe. In addition, a 52-week placebo-controlled trial of evolocumab for hyperlipidemia found that patients in the evolocumab group had a 58% reduction in LDL-C at week 12 and a 57% reduction at week 52 compared with the placebo group [83]. The long-term safety and tolerability of alirocumab were evaluated, and the average reduction in LDL-C at week 78 was 58% [54]. These results indicate that the LDL-C reduction effect of PCSK9 inhibitors is durable and well tolerated.

### 4.2. Atherosclerosis

As is a chronic inflammatory vascular disease caused by vascular endothelial injury, which is characterized by the gradual accumulation of plaque in the walls of large and medium-sized arteries. Commonly used drugs such as antiplatelet, anticoagulant and vasodilator drugs are used to mainly slow the accumulation of plaque by lowering blood lipids to induce thrombolysis, and prevent vascular blockage, respectively. Several studies have shown that PCSK9 inhibitors have certain effects on inhibiting vascular endothelial cell injury, lowering blood lipids, and protecting cardiac cells during myocardial ischemia/reperfusion (I/R).

As a class of lipid-lowering drugs, PCSK9 inhibitors can effectively slow the accumulation of intravascular plaque and even cause plaque regression. Zeng et al. [84] found that PCSK9 inhibitors in human umbilical vein endothelial cells induce pyroapoptosis by mediating the expression of oxidized LDL through the mitochondrial ubiquinone-cytochrome C reductase core protein 1/reactive oxygen species pathway. These results suggest that PCSK9 inhibitors can inhibit scortosis of endothelial cells to some extent and slow down the development of As. In addition, Leila Safaeian et al. [55] showed that evolocumab had antioxidant and cellular protective effects on H_2_O_2_-induced oxidative damage in endothelial cells. Nicholls et al. conducted a multicenter, double-blind, placebo-controlled randomized clinical trial [85] in which evolocumab or placebo was injected once a month (420 mg) and measured the atherosclerotic plaque volume percentage (PAV) by intravascular ultrasound as the primary focus during 84 weeks of treatment. Angiogram data from 968 patients with coronary heart disease were collected and showed lower LDL-C levels in the evolocumab group than in the placebo group. In addition, those treated with statins had a 0.05% increase in PAV while taking the placebo and a 0.95% decrease while taking evolocumab. Among those not taking statins, 64.3% of patients in the evolocumab group had plaque regression, compared with 47.3% in the placebo group. There are few reports on the role of PCSK9 in vascular injury. Arianna Toscano et al. enrolled patients with HeFH and performed biochemical analysis and PWV assessment at baseline (T0) for 6 months of HEstatin plus ezetimibe (T1) and 6 months of PCSK9-I (T2). PCSK9 levels increased during statin/EZE treatment (+42.8% at T1) and began to decrease after treatment (−34.4% at T2). It is suggested that PCSK9 levels are related to baseline PWV values in the HeFH discipline. In addition, we found that changes in PCSK9 levels seemed to correlate with PWV image changes [86]. These results suggest that PCSK9 inhibitors can significantly slow down, stabilize and even decrease plaques and have a synergistic effect with statins.

Acute myocardial ischemia is one of the main causes of As plaque shedding. In the process of acute myocardial ischemia, the level of PCSK9 increases, leading to the deterioration of the myocardial inflammatory microenvironment and cardiac insufficiency. Therefore, PCSK9 inhibitors may have a protective effect on myocardial injury. Through animal experiments, Siripong Palee et al. [87] found that PCSK9 inhibitor administration before ischemia had a cardioprotective effect and improved left ventricular function by alleviating mitochondrial damage, which was demonstrated by reducing myocardial infarction size and arrhythmia at I/R.

## 5. Potential Therapeutic Applications of PCSK9

Efficacy in the cardiovascular field is not the only role of PCSK9 inhibitors. Recent studies have found that PCSK9 inhibitors have potential application value in the treatment of sepsis, some tumors, some viral infections, and other diseases. In this section, we summarize their application (Figure 2 and Table A1 in Appendix A).

### 5.1. Application of PCSK9 in the Treatment of Sepsis

PCSK9 reduces circulating cholesterol by reducing the density of LDL-Rs on liver cells. During infection, these receptors are involved in clearing circulating bacterial lipids such as LPS. These lipids are thought to play a key role in initiating uncontrolled systemic inflammatory responses during sepsis [88]. Lipid clearance by inhibiting PCSK9 may be a new method for the treatment of sepsis [20]. It has been reported that PCSK9 KO mice have a protective effect on LPS-induced septic shock [89], and the PCSK9 dysfunction type (LOF) variant has been reported to show low frequencies of septic shock and organ failure [90], whereas the reverse was observed in transgenic mice overexpressing PCSK9 [91]. In sepsis models of cecal ligation and perforation, PCSK9^−/−^ mice have indeed been reported to have lower bacterial concentrations in the blood, lungs, and peritoneal fluid than wild-type animals, suggesting that deletion of PCSK9 is beneficial for bacterial inhibition or clearance [91]. As the septic protection of PCSK9^−/−^ was not shown in LDL-R KO mice, KO of PCSK9 or the use of PCSK9 inhibitors can enhance lipid clearance through LDL-R [92,93].

Notably, the complexity of sepsis is species-specific, casting doubt on the broad use of rodent models of sepsis [94]. In fact, mice and rats are significantly more resistant to sepsis than humans. The humanized mouse model generated by Laudanski K. et al. alleviated this limitation to a certain extent [95]. In addition, PCSK9 inhibitors may not benefit young hosts with sepsis. Mihir R. Atreya et al. found an enhanced association between PCSK9 LOF mutations and poor prognosis of septic shock in children [96]. The rationale behind this observation is unclear, and they suggest that children should be excluded from sepsis clinical trials involving PCSK9 inhibitors until the unknown effect of PCSK9 dysfunction in young children is clarified.

Vecchie A. et al. found that in their study on the correlation between PCSK9 and septic shock mortality, patients with septic shock with a low level of PCSK9 on the first day showed higher 28- and 90-day mortality than other patients with a high level of PCSK9 on the first day [97]. In the subanalysis of this experiment, researchers found that low circulating PCSK9 levels one day after the onset of sepsis were not associated with a good prognosis [98,99]. In addition, patients with low high-density lipoprotein (HDL) during sepsis have a much greater risk of death from organ failure [100], whereas PCSK9 has no significant effect on the known critical HDL levels [101]. Therefore, the use of PCSK9 inhibitors during sepsis is unknown, and the preventive effect of PCSK9 inhibitors on sepsis cannot be ruled out.

### 5.2. Application of PCSK9 in Tumor Diagnosis and Treatment

Cholesterol plays a key role in cell metabolism, especially in energy-demanding processes such as cell growth and division. Tumor cells have higher requirements for cholesterol. LDL provides cholesterol to peripheral cells, and its upregulation in tumors is related to the progression of cancer [102]. It has been reported that in different patient cohorts, individuals with high expression of tumor PCSK9 mRNA have a poorer overall survival rate than individuals with low expression of PCSK9 mRNA [21]. Therefore, PCSK9 expression may be a valuable biomarker for the clinical prognosis of some malignant tumors, including liver, stomach, kidney, pancreatic, and breast cancers [103].

It has been shown that in CD8^+^ T cells, LDL-R forms a complex with the T-cell receptor (TCR), which is activated by binding to antigenic peptides presented to tumor cells by the major histocompatibility complex (MHC). The binding of LDL-R to TCR promotes cell surface recycling and enhances the antitumor activity of CD8^+^ T cells [104]. Therefore, the combination of cholesterol-lowering drugs (PCSK9i, statins, and ezetimibe) can increase the level of surface LDL-R and significantly reduce LDL-C. Reducing tumor growth and/or metastasis through cholesterol depletion and enhancing TCR and MHC-I activity is an important strategy for cancer treatment. Liu et al. [21] found that the tumorigenicity of pcSK9-KO tumor cells was reduced in mice, and programmed death receptor 1 (PD-1) mAb was injected to inhibit tumor growth in conjunction with PCSK9 inhibitors, suggesting that inhibition of tumor-derived PCSK9 can overcome drug resistance to PD-1 therapy.

### 5.3. Application of PCSK9 in Viral Infection

PCSK9 may be associated with viral infectious diseases, including hepatitis C virus (HCV), dengue fever virus (DENV), and SARS-CoV-2, the etiological agent of COVID-19. LDL-R has been identified as one of the binding receptors for HCV entry into liver cells [100]. Caron J. et al. [22] isolated iPSCs from a patient with LDL-R deletion, enabling them to induce differentiation into liver cells. Cells lacking functional LDL-R can still be infected with HCV, but virus production is significantly increased after LDL-R reexpression, suggesting that LDL-R is not related to HCV entry but is closely related to the lipid metabolism of host cells during HCV packaging [22]. Because PCSK9 inhibitors can enhance HCV packaging and infectivity, patients infected with HCV should use PCSK9 inhibitors with caution [22].

In addition, studies have found that DENV infection promotes cholesterol synthesis through the SREBP-2 pathway [105], induces the expression of PCSK9 in hepatocytes [23], and reduces circulating LDL-C. Elevated plasma PCSK9 levels have been detected in patients infected with DENV, and elevated cholesterol levels in the endoplasmic reticulum lead to a significantly reduced antiviral type I interferon (IFN) response in host hepatocytes, high levels of viremia, and more severe plasma leakage [22]. Therefore, PCSK9 inhibitors may benefit DENV patients by increasing the levels of antiviral interferon response genes [22]. Combination therapy with statins and PCSK9 inhibitors has the potential to reduce two factors that contribute to an increased risk of complications from COVID-19 thrombosis. Therefore, the use of PCSK9 inhibitors in the treatment of COVID-19 may be another potential strategy.

## 6. Summary and Outlook

By reducing LDL-C content, inhibiting PA and thrombosis, and reducing the production of inflammatory substances, PCSK9 can treat hyperlipidemia [106], As [107], lipoprotein(a) [108], venous thromboembolism (VTE) risk [109], aortic stenosis [110] and other cardiovascular diseases. In addition, PCSK9 inhibitors have been used to improve the treatment of sepsis [92] and some malignant tumors [103,111,112]. Existing PCSK9 inhibitors include monoclonal antibodies, small interfering RNAs or ASOs, small molecule drugs, polypeptide mimics, gene editing techniques, and vaccines. Currently, the inhibitors in clinical use include monoclonal antibodies and siRNA.

Fully human monoclonal antibodies, alirocumab (Regeneron and Sanofi) and evocumab (Amgen), approved by the FDA, are primarily used for HeFH and the prevention of CV events in patients with confirmed cardiovascular disease [33]. Evocumab has significant efficacy and good tolerability for HoFH [25,113,114] but has no obvious efficacy for homozygous FH patients with negative LDL-R receptor defect genes who require additional treatment strategies [107,113]. The adverse reactions of antibodies were not significantly different from placebo, except that the response rate at the injection site was slightly higher than that of placebo [107].

Currently available PCSK9 inhibitors include alirocumab and evolocumab, as well as inclisiran, a siRNA drug. PCSK9 monoclonal antibody does not have the same biological effect as the PCSK9 variant that reduces LDL-C levels by binding to free PCSK9 protein [115]. In addition, the production of recombinant mAb is complicated, the cost is high, and the preservation conditions are strict. Patients need to spend a lot of time in therapy and bear high economic costs [25,107], which brings great limitations to clinical use.

N-acetylgalactosamine and restricted ethyl-modified ASOs are novel therapies for inhibiting PCSK9. In addition to injection, American researchers have also developed an ASO drug (AZD8233) with oral potential. Studies have found that both subcutaneous injection and oral administration can significantly reduce the level of PCSK9, and compared with subcutaneous injection, oral drugs are more convenient for patients to use, avoid the potential risks and discomfort of subcutaneous injection, and improve patient compliance [57]. However, the current clinical data of this drug are not sufficient, and the subsequent effectiveness and safety need to be confirmed in more clinical studies.

SiRNA drugs or ASOs have reduced the frequency of administration or injection but still require repeated treatment. Vaccines are an effective way to reduce these limitations. PCSK9 vaccine achieves therapeutic effects by triggering the production of host anti-PCSK9 antibodies and appropriately neutralizing PCSK9 and LDL-R interactions [72]. Some studies have modified the immunogenic peptide constructs of IFPT onto L-IFPT and then mixed them into alum adjuvant, naming them L-IFPTA + vaccine [73]. Studies have shown that the antibodies induced by this method are durable [74] and have the potential to be an effective vaccine candidate for the treatment of dyslipidemia and atherosclerotic cardiovascular disease, but clinical data for L-IFPTA are still lacking.

Meanwhile, the immuno safety and efficacy of the vaccine in inducing anti-PCSK9 antibody responses in nonhuman primates are being investigated. Although the PCSK9 vaccine development path still requires extensive clinical studies, the available safety and efficacy evidence is encouraging. In addition to PCSK9 vaccination, gene editing technology is also a new technique to inhibit the function of PCSK9. The CRISPR adenine base editor is delivered by LNPs, which can effectively and accurately modify relevant genes in vivo and effectively reduce PCSK9 protein and LDL-C levels, but it is still in the stage of phase I clinical trials [78,79,116]. In summary, PCSK9 inhibitors significantly reduce the level of LDL-C, and the development of related drugs is of great significance for the treatment of cardiovascular diseases.

## Figures and Tables

**Figure 1 cells-11-02972-f001:**
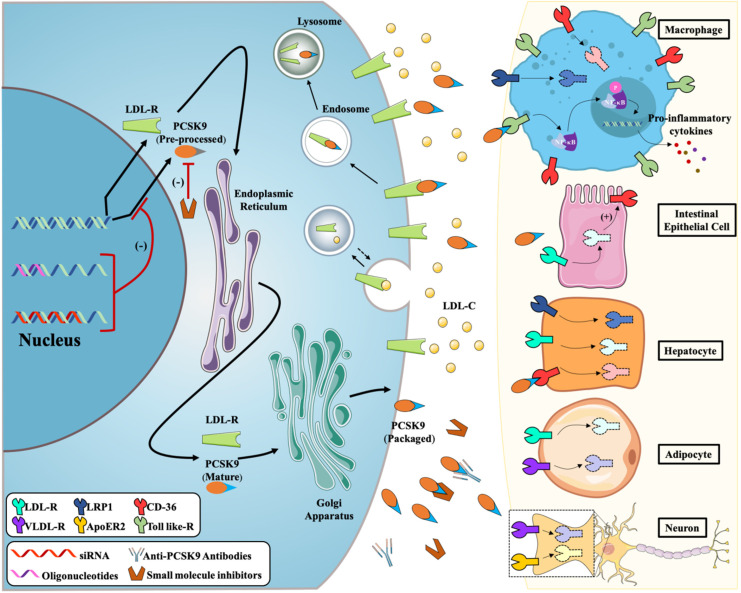
Molecular mechanisms of PCSK9 inhibitors with different strategies.

**Figure 2 cells-11-02972-f002:**
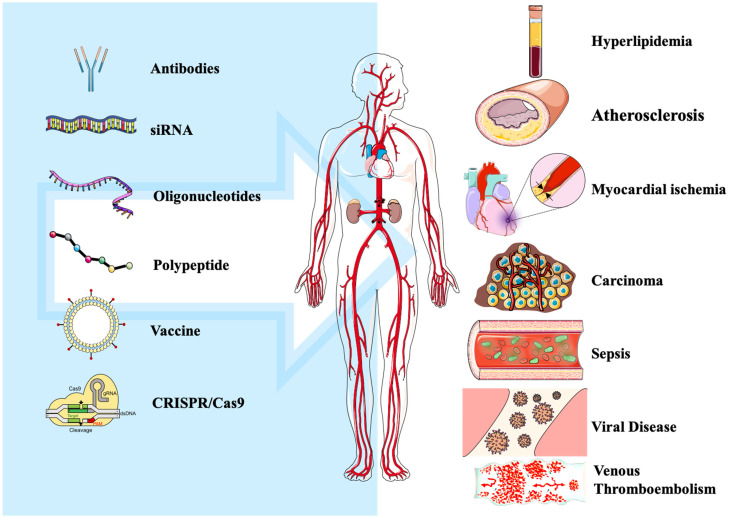
Application of PCSK9 inhibitors with different strategies in disease.

**Table 1 cells-11-02972-t001:** Application of PCSK9 inhibitors with different strategies in disease.

State	Drug	Strategies	Application	Security	Mechanism	Ref.
Marketed drugs	Alirocumab	Monoclonal antibody	Cardiovascular disease and primary hyperlipidemia, including HeFH and HoFH	Safe, efficient, with occasional adverse reactions	PCSK9 binds to LDL-R on the surface of liver cells, promoting the degradation of LDL-R in the liver and reducing LDL-C levels.	[27,54]
	Evolocumab	Monoclonal antibody	Cardiovascular disease and primary hyperlipidemia, including HeFH and HoFH	Safe, efficient, with occasional adverse reactions	PCSK9 binds LDL-R, degrades LDL-R, and removes LDL-C.	[27,55]
	Inclisiran	siRNA	Adults with HeFH or clinical ASCVD in patients who require additional lowering of LDL-C	Safe, with occasional injection site adverse reactions	Inclisiran interferes with PCSK9 in liver cells by RNA, increases the circulation and expression of LDL-C receptors on the surface of liver cells, the uptake of LDL-C, and the reduction of LDL-C.	[26,56]
Drugs under development	AZD8233	ASO	Hyperlipidemia	Still in clinical trial	It is designed to target PCSK9 mRNA, thereby inhibiting the intracellular protein translation and synthesis of PCSK9 protein. Targets endogenous liver Pcsk9 expression with a GalNAc-ASO approach to ablate circulating PCSK9, induce hepatic LDL-R expression, and thereby reduce plasma total and LDL-C concentrations.	[57]
	MiR-552-3p	ASO	Hyperlipidemia	To be tested	MiR-552-3p can bind to the 3′ untranslated region (3′-UTR) of PCSK9 to inhibit translation and interact with the promoter of PCSK9 to suppress transcription and increase the LDL-R protein level, promote LDL-C uptake, and lower serum LDL-C.	[58]
	BMS-962476	Antibody-like protein	Hypercholesterolemia	The experimental data are not sufficient	BMS-962476 blocks PCSK9 biologic activity by preventing binding and cointernalization with LDL-R during endocytosis, resulting in interruption of the subsequent sorting/degradation steps and increased receptor recycling and LDL uptake.	[52]
	L-IFPTA vaccine	Vaccine	Hypercholesterolemia and As	Long-lasting, durable, and safe	L-IFPTA vaccine-induced functional antibodies can specifically bind to circulating PCSK9, inhibit its interaction with LDL-R and thereby increase the expression of LDL-R on the surface of liver cells, leading to significant reductions in TC and (V)LDL-C.	[59,60]
	Annexin A2	Polypeptide	Hyperlipidemia	To be tested	Annexin A2 interacts with the M1 and M2 Domains of PCSK9; the function of PCSK9 induced LDL-R was blocked, and PCSK9 was inhibited.	[57,61]
	7030B-C5	Small molecule	Cardiovascular disease	To be tested	HNF1 α and FoxO3 regulate transcription to inhibit PCSK9 expression and increase HepG2 cell-mediated total LDL-R protein and its uptake of LDL-C.	[62]
	Pep2-8	Polypeptide	Cardiovascular disease	To be tested	Neutralize the activity of PCSK9 and realize the functional recovery of cellular LDL receptors.	[63,64]
	MK-0616		Hypercholesterolemia	Still in clinical trial	Mk-0616 interferes with the binding of LDL-C to LDL-R and causes the liver to express more LDL-R, thereby reducing plasma LDL-C levels.	[65,66]

Abbreviations: PCSK9, protein convertase subtilisin/kexin type 9; LDL-R, low-density lipoprotein receptor; LDL-C, LDL cholesterol; HeFH, heterozygous familial hypercholesterolemia; HoFH, homozygous familial hypercholesterolemia; As, atherosclerosis; ASCVD, atherosclerotic cardiovascular disease; ASO, antisense oligonucleotides; GalNAc, N-acetylgalactosamine; L-IFPT, liposome nanoparticles; HNF1 α, hepatocyte nuclear factor 1A; FoxO3, forkhead box protein O3; HepG2, human laryngeal carcinoma epithelial cells; Pep2-8, Pep2-8 is a PCSK9 inhibitor.

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
