# Peer review of "PCSK9 Inhibition: From Current Advances to Evolving Future"

_cells, 2022, doi:10.3390/cells11192972_

Round 1

Author Response

Response to Reviewer 1 Comments

Reviewer’s comments for: “PCSK9 Inhibition: From Current Advances to Evolving Future” by Liu C et al.

Point 1: Line 37: Authors indicate that PCSK9 activates NF-kB and NLRP3 inflammasome promoting inflammation independent of its effect on circulating LDL. In humans, however, this effect of PCSK9 does not exist independently of that on circulating LDL and tissue uptake of LDL. Subjects with low plasma PCSK9 were found to have higher LDLR and CD36 on the adipose tissue surface and higher adipose tissue NLRP3 inflammasome activation and secretion of IL-1B (Cyr Y 2021 Physiol Rep). They also had higher risk factors for diabetes. This human finding and others related to higher inflammasome priming and/or activation with low plasma PCSK9 or PCSK9 LOF variants (Gagnon A et al 2016 Obesity) are not mentioned. How do authors view these findings for a more balanced model of PCSK9 action in vivo? 

Response: Thank you very much for your comments. We apologize for the confusion caused by our expression. The expression "independent of its effect on circulating LDL" is not rigorous and has been changed to "not entirely dependent" on LDLR in the new manuscript. NF-kb and NLRP3 inflammasomes are more closely related to Toll-like receptors than CD36 and LDLR. In addition to promoting inflammation by activating NF-KB and NLRP3 inflammasomes through LDLR, PCSK9 also acts through TLR4 receptors [1-3].

Reference:

[1]Tang, Z.; Peng, J.; Ren, Z.; Yang, J.; Li, T.; Li, T.; Wang, Z.; Wei, D.; Liu, L.; Zheng, X.; et al. New role of PCSK9 in atherosclerotic inflammation promotion involving the TLR4/NF-κB pathway. Atherosclerosis 2017, 262, 113-122, doi:10.1016/j.atherosclerosis.2017.04.023.

[2]Pedrinelli, R. PCSK9 Induces Tissue Factor Expression by Activation of TLR4/NFkB Signaling. International Journal of Molecular Sciences 2021, 22.

[3]Yang, C.; Zeng, Y.; Hu, Z.; Liang, H. PCSK9 promotes the secretion of pro-inflammatory cytokines by macrophages to aggravate H/R-induced cardiomyocyte injury by activating NF-κB signaling. General physiology and biophysics 2020, 39, 123-134, doi:10.4149/gpb-2019057.

Point 2:  Sentence on line 559 is an understatement “although the reduction in cardiovascular risk with PCSK9 inhibitors should far outweigh any potential increase in diabetes risk”. Diabetes increases the risk of CVD by 2-4 times; thus, developing diabetes on hypocholesterolemic agents (already shown for statins) defeats the purpose of these drugs. 

Response: Thank you for pointing out these problems. In the revised manuscript, we have revised this section.

Point 3:  References for line 37 related to PCSK9 discovery is missing 

Response: Thank you for your careful review. In the revised manuscript, we have added the references.

Point 4:  Paragraph Line 182 – 194 is a duplicate of paragraph 195-206

Response: Thank you for your careful review. In the revised manuscript, we have deleted the duplicate paragraph.

Point 5:The sentence starting line 247 is unclear/incomplete: (4) Inhibition of circulating PCSK9 activity: PCSK9 vaccine et al”. 

Response: Thank you very much for pointing out this problem. We have revised the sentence.

Point 6: This review addresses the advancement in PCSK9 inhibition therapy and discusses its use in human disease (hyperlipidemia, CVD, sepsis, cancer, infection).

- There are multiple reviews in the literature that fully cover this topic, the latest in 2022 in Endocrine Reviews being “The Multifaceted Biology of PCSK9” by Nabil Seidah and Annik Prat, the lab that discovered PCSK9 in 2003. Thus, the novelty of this review and its contribution to the field are unclear.

Response: Thank you very much for your comments. It is well known that LDLR is one of the most important receptors of PCSK9, but there are few reports about other related receptors. In this review, a series of mechanisms related to PCSK9 are summarized. In addition to LDLR, mechanisms including Toll-like receptors closely related to inflammation and VLDL-R and ApoER2 receptors related to lipid metabolism were more systematically summarized. In addition, unlike "The Multifaceted Biology of PCSK9", we focus more on the induction of various drug forms, including monoclonal antibodies, peptides, small molecules, and CRISPR‒Cas9. In conclusion, our review systematically summarizes PCSK9 from another dimension, which we believe will help readers understand the mechanism of action and drug progress of PCSK9 more comprehensively.

Reviewer 2 Report

On page 2 and page 7, the authors state that PCSK9 inhibition is associated with cognitive impairment. This is not true and has been disproved by randomized clinical trials. There were no cognition issues is FOURIER, ODYSSEY, and the dedicated Ebbinghaus study confirmed this as well. Based on these trials, very low LDL-C is safe and not associated with cognitive issues. The author should share this trial data if they want to cover the topic of cognition thoroughly.

Page 6 has confusing terminology while grouping PCSK9 drugs into 4 groups. The first group is "PCSK9 inhibitors" but it does not list monoclonal antibodies until group 3. I would restructure this paragraph.

Also on page 6 (group 4), vaccines, et al is listed. It is unclear what the "et al" is referring to, so I would either state additional therapies or leave it at "vaccines".

On page 7, the Rutherford-2 data test evolocumab, not Bocacizumab.

In discussing small-molecule drugs, what about MK-0616 from Merck? And should add to the summary table.

Page 8 under vaccine development, the authors state that PCSK9 inhibitors have had “many clinical limitations”, which is not true. These drugs are incredibly well tolerated and do not have any significant side effects. The authors go on to state that there is a possibility of inducing anti-monocloncal antibodies in the host. This was true for bococizumab which was not FDA approved for that reason, but it is inaccurate to say that is a clinical issue for the available PCSK9 inhibitors. The authors should remove these sentences and not use this as a reason for the vaccine. The benefit of a vaccine is that it is one and done. So q2 week injections are no longer needed and compliance is not an issue.

A CRISPR PCSK9 trial is now in human trials and has been dosed in the first patient. The CRISPR section should be updated accordingly.

Section 4.1:

Beta blockers are not used to treat lipid disorders. Would remove this.

Would add bempedoic acid to this list.

PCSK9 inhibitors have been FDA approved for 7 years so would not call them new.

Eluzamab is a typo.

Additional points worth mentioning:

PCSK9 monoclonal antibodies significantly lower Lp(a) (O’Donoghue, Circulation)

PCSK9 monoclonal antibodies lower VTE risk (Marston, Circulation)

PCSK9 monoclonal antibodies slow aortic stenosis progression (Bergmark, Circulation)

VTE and AS should be added to Figure 2 for a more complete picture of the benefits.

There has been no increased risk of diabetes in the PCSK9 inhibitor outcomes trials. I don’t think this meets the bar for being in the summary section.

In the summary, the authors again bring up anti-drug antibodies with PCSK9 inhibitors:  “The high titer of anti-drug antibodies produced in patients decreases rapidly over 566 time, requiring frequent dosing". This is false and needs to be removed.

In the summary the authors argue that small molecule PCSK9 inhibitor are lower in price and have better stability. I don’t think these comments can be made as they are still in early clinical trials and a final drug has not been created or given a price point.

I found the Summary to be much too long (9 paragraphs) and very repetitive from earlier text. I would shorten considerably and just stick to the key takeways.

The final paragraph is more fitting for a high-level summary, however to say we don’t have long-term safety data for alirocumab and evolocumab is inaccurate. OSLER-1 has published 5-year follow up data and the drug has been FDA approved for 7 years. There have been no safety concerns identified. I don't think the main takeaway of this broad review article is that we should be concerned about long-term safety.

Author Response

Response to Reviewer 2 Comments

Point 1: On page 2 and page 7, the authors state that PCSK9 inhibition is associated with cognitive impairment. This is not true and has been disproved by randomized clinical trials. There were no cognition issues is FOURIER, ODYSSEY, and the dedicated Ebbinghaus study confirmed this as well. Based on these trials, very low LDL-C is safe and not associated with cognitive issues. The author should share this trial data if they want to cover the topic of cognition thoroughly.

Response: Thank you for pointing out this problem. We have discussed this section and described the different opinions. (lines 74-76)

Point 2: Page 6 has confusing terminology while grouping PCSK9 drugs into 4 groups. The first group is "PCSK9 inhibitors" but it does not list monoclonal antibodies until group 3. I would restructure this paragraph.

Response: Thank you for pointing out this problem. We apologize for our misdescription. In the revised manuscript, we have modified it. (lines 238-255)

Point 3: Additionally, on page 6 (group 4), vaccines, et al is listed. It is unclear what the "et al" is referring to, so I would either state additional therapies or leave it at "vaccines".

Response: Thank you for pointing out this problem. We have revised this problem in the new manuscript and checked similar problems in the full text. (line254-255)

Point 4: On page 7, the Rutherford-2 data test evolocumab, not Bocacizumab.

Response: Thank you very much for pointing out this problem. We have revised this problem in the new manuscript and checked similar problems in the full text. (lines 266-269)

Point 5: In discussing small-molecule drugs, what about MK-0616 from Merck? And should add to the summary table.

Response: Thank you for your suggestion. We have added relevant content to the summary table in the revised manuscript. (The last line of Table 1)

Point 6: Page 8 under vaccine development, the authors state that PCSK9 inhibitors have had “many clinical limitations”, which is not true. These drugs are incredibly well tolerated and do not have any significant side effects. The authors go on to state that there is a possibility of inducing anti-monocloncal antibodies in the host. This was true for bococizumab, which was not FDA approved for that reason, but it is inaccurate to say that is a clinical issue for the available PCSK9 inhibitors. The authors should remove these sentences and not use this as a reason for the vaccine. The benefit of a vaccine is that it is one and done. Therefore, q2 week injections are no longer needed, and compliance is not an issue.

Response: Thank you for pointing out this problem. We have revised it in the new manuscript. (lines 345-349)

Point 7: A CRISPR PCSK9 trial is now in human trials and has been dosed in the first patient. The CRISPR section should be updated accordingly.

Response: Thank you for pointing out this problem. We have updated the CRISPR PCSK9 trial of the first patient in the new manuscript. (lines 384-390)

Point 8: Section 4.1:

Beta blockers are not used to treat lipid disorders. Would remove this.

Would add bempedoic acid to this list.

Response: Thank you for pointing out these problems. In the revised manuscript, we have revised the tenses in this section and checked the similar problems in the full text. In addition, we have added bempedoic acid to this list. (lines 400-401)

Point 9: PCSK9 inhibitors have been FDA approved for 7 years so would not call them new.

Eluzamab is a typo.

Response: Thank you for pointing out these problems. In the revised manuscript, we have revised “New drug” to “new dosage form design”.

Point 10: Additional points worth mentioning:

PCSK9 monoclonal antibodies significantly lower Lp(a) (O’Donoghue, Circulation)

PCSK9 monoclonal antibodies lower VTE risk (Marston, Circulation)

PCSK9 monoclonal antibodies slow aortic stenosis progression (Bergmark, Circulation)

Response: Thank you very much for your comments. In the revised manuscript, we have added relevant content. (line564)

Point 11: VTE and AS should be added to Figure 2 for a more complete picture of the benefits.

Response: Thank you very much for your suggestion. We have added VTE and AS to Figure 2 in the revised manuscript. (page 14)

Point 12: There has been no increased risk of diabetes in the PCSK9 inhibitor outcomes trials. I don’t think this meets the bar for being in the summary section.

Response: Thank you for pointing out this problem. There are some incorrect expressions in our manuscript, and we have corrected them in the revised manuscript. In addition, we have checked and corrected similar problems in the full text.

Point 13:In the summary, the authors again bring up anti-drug antibodies with PCSK9 inhibitors: “The high titer of anti-drug antibodies produced in patients decreases rapidly over 566 time, requiring frequent dosing". This is false and needs to be removed.

Response: Thank you for pointing out this problem. We have deleted “The high titer of anti-drug antibodies produced in patients decreases rapidly over time, requiring frequent dosing" in the revised manuscript and checked and corrected similar problems in the full text.

Point 14:In the summary the authors argue that small molecule PCSK9 inhibitor are lower in price and have better stability. I don’t think these comments can be made as they are still in early clinical trials and a final drug has not been created or given a price point.

 Response: Thank you very much for pointing out this problem. We have deleted the comments about small molecule PCSK9 inhibitors in the revised manuscript and checked and corrected similar problems in the full text.

Point 15: I found the Summary to be much too long (9 paragraphs) and very repetitive from earlier text. I would shorten considerably and just stick to the key takeways.

Response: Thank you for your suggestions. In view of the problems in the summary, we have condensed this section.

Point 16: The final paragraph is more fitting for a high-level summary, however to say we don’t have long-term safety data for alirocumab and evolocumab is inaccurate. OSLER-1 has published 5-year follow-up data, and the drug has been FDA approved for 7 years. There were no safety concerns identified. I do not think the main takeaway of this broad review article is that we should be concerned about long-term safety.

Response: Thank you for pointing out this problem. In view of the problems in "long-term safety data for alirocumab and evolocumab", we have deleted the comments in the revised manuscript.

Reviewer 3 Report

This is an interesting review about the role of PCSK9 on several pathological pathways and current and future therapies against PCSK9. The review is interesting, well performed and the english style is adequate. I only have minor suggestions for the authors:

- Please perform a paragraph about the role of PCSK9 mAb on platelet activity and thrombosis; also, please consider this recent paper about this pathological pathway (10.3390/ijms22137193) and comment this in the new paragraph.

- In the paragraph "Atherosclerosis", the authors could include the role of PCSK9 and its inhibition on arterial stiffness; please consider this manuscript (10.3390/biom12040562) and comment this in the paragraph

Author Response

Response to Reviewer 3 Comments

This is an interesting review about the role of PCSK9 on several pathological pathways and current and future therapies against PCSK9. The review is interesting, well performed and the english style is adequate. I only have minor suggestions for the authors:

Point 1: Please perform a paragraph about the role of PCSK9 mAb on platelet activity and thrombosis; also, please consider this recent paper about this pathological pathway (10.3390/ijms22137193) and comment this in the new paragraph.

Response:Thank you very much for your suggestions. We have performed a paragraph about the role of PCSK9 mAb on platelet activity and thrombosis. (line157-166)

Point 2: In the paragraph "Atherosclerosis", the authors could include the role of PCSK9 and its inhibition on arterial stiffness; please consider this manuscript (10.3390/biom12040562) and comment this in the paragraph.

Response:Thank you very much for your suggestions. We have supplemented the relevant literature (10.3390/biom12040562). (line454-462)

Round 2

Reviewer 2 Report

The authors have addressed the my comments and I just have the following minor comments:

I don't understand the phrase "new dosage form design". Would suggest changing.

The classification of PCSK9 inhibitors is still confusing. It now reads nucleic acids for both #2 and #3. I think #3 should be small molecule.

he header for 4.2 appears to be a typo.

Author Response

The authors have addressed my comments and I just have the following minor comments:

I don't understand the phrase "new dosage form design". Would suggest changing.

Response: Thank you for your valuable comments.Your suggestions are very professional. In the revised manuscript, we have changed the “new dosage Form Design” to “drug research and development”.

compressed the citation of review articles and increased the citation of research articles.

Response: Thank you for your valuable comments. We are deeply aware of these problems. In the revised manuscript, we have compressed the citation of review articles and increased the citation of research articles.

The classification of PCSK9 inhibitors is still confusing. It now reads nucleic acids for both #2 and #3. I think #3 should be small molecule.

Response: Thank you very much for your comments. We are sorry for our misdescription. And we have revised the points in the new manuscript.

he header for 4.2 appears to be a typo.

Response: Thank you for pointing out the problem. We haverevisedit in the new manuscript.